# Production of H_2_-Free Carbon Monoxide from Formic Acid Dehydration: The Catalytic Role of Acid Sites in Sulfated Zirconia

**DOI:** 10.3390/nano12173036

**Published:** 2022-09-01

**Authors:** Hyun Ju Lee, Dong-Chang Kang, Eun-Jeong Kim, Young-Woong Suh, Dong-Pyo Kim, Haksoo Han, Hyung-Ki Min

**Affiliations:** 1Department of Chemical and Biomolecular Engineering, Yonsei University, Seoul 03722, Korea; 2Department of Chemical Engineering, Pohang University of Science and Technology (POSTECH), Pohang 37673, Korea; 3School of Energy and Chemical Engineering, Ulsan National Institute of Science and Technology (UNIST), Ulsan 44919, Korea; 4Department of Chemical Engineering, Hanyang University, Seoul 04673, Korea; 5LOTTE Chemical Research Institute, Daejeon 34110, Korea

**Keywords:** sulfated zirconia, formic acid, acidity, dehydration, carbon monoxide

## Abstract

The formic acid (CH_2_O_2_) decomposition over sulfated zirconia (SZ) catalysts prepared under different synthesis conditions, such as calcination temperature (500–650 °C) and sulfate loading (0–20 wt.%), was investigated. Three sulfate species (tridentate, bridging bidentate, and pyrosulfate) on the SZ catalysts were characterized by using temperature-programmed decomposition (TPDE), Fourier-transform infrared spectroscopy (FTIR), and X-ray photoelectron spectroscopy (XPS). The acidic properties of the SZ catalysts were investigated by the temperature-programmed desorption of *iso*-propanol (IPA-TPD) and pyridine-adsorbed infrared (Py-IR) spectroscopy and correlated with their catalytic properties in formic acid decomposition. The relative contributions of Brønsted and Lewis acid sites to the formic acid dehydration were compared, and optimal synthetic conditions, such as calcination temperature and sulfate loading, were proposed.

## 1. Introduction

High-purity carbon monoxide (CO) is an important industrial dry etching gas used for semiconductor production [1]. Generally, CO is obtained from the syngas production processes (i.e., steam methane reforming (SMR) and partial oxidation (POX) of hydrocarbons) combined with various separation technologies, such as cryogenic separation, pressure swing adsorption (PSA), and membrane separation. However, producing high-purity CO (>99.995%) by using the aforementioned technologies is practically difficult, owing to the presence of gaseous impurities in syngas, such as methane (CH_4_), nitrogen (N_2_), carbon dioxide (CO_2_), oxygen (O_2_), and moisture, which cannot be easily separated [2]. In addition, the formation of volatile metal carbonyl impurities (Fe(CO)_5_ and Ni(CO)_4_) with ppb-level under high temperature and pressure gives detrimental effects on the quality of semiconductor products. Decomposition of refined formic acid (CH_2_O_2_) to CO and H_2_O is an alternative to producing high-purity CO, and this process does not involve interference from impurity gases (i.e., CH_4_, N_2_, and O_2_) and formation of metal carbonyls [3].

Depending on the type of catalyst, the catalytic decomposition of formic acid proceeds via two different routes: dehydration (Equation (1)) to CO and H_2_O over acidic catalysts, and dehydrogenation (Equation (2)) to H_2_ and CO_2_ over metal or basic catalysts [4,5,6]. To date, dehydrogenation of formic acid has been more intensively studied than dehydration as a means of hydrogen storage and carriers [7,8,9,10]. However, the H_2_-free dehydration of formic acid in Equation (1) is essential for producing high-purity CO, whereas with regard to Equation (2), producing CO_2_ and H_2_ is an undesirable reaction that must be suppressed [11]. In the dehydration pathway, the conversion of formic acid generally increases proportionally to the concentration of Brønsted acid sites in the catalysts [12]:HCOOH ↔ CO + H_2_O (△H°_298_ = 29.20 kJ mol^−1^)(1)
HCOOH ↔ CO_2_ + H_2_ (△H°_298_ = 31.20 kJ mol^−1^)(2)

Zirconia (ZrO_2_) is a unique metal oxide that is widely used as a catalyst or support in various applications, owing to its excellent thermal stability and controllable acidic and basic properties [13,14,15,16]. Although pure ZrO_2_ is Lewis acidic, many attempts have been made to increase its Brønsted acidity, such as the addition of acidic metal oxides and sulfate ions. As reported by Lee et al., the addition of WO_3_ enhances the Brønsted acidity of ZrO_2_, resulting in an increase in the catalytic activity for formic acid dehydration [17]. Oki et al. successfully increased the Brønsted acidity of ZrO_2_ by introducing MoO_3_ and obtained high catalytic activity for the polyesterification of adipic acid with 1,4-butanediol [18]. In the Cr_2_O_3_-ZrO_2_ system, the nature of the acid sites can be controlled by applying different precursors; that is, Cr_2_O_3_ from ammonium chromate generates Brønsted acid sites, while chromium nitrate leads to the formation of Lewis acid sites [19]. Sulfated zirconia (SZ), which is activated by various sulfating agents, such as H_2_SO_4_, (NH_4_)_2_SO_4_, and H_2_S, is a class of solid superacids exhibiting outstanding catalytic performance for a variety of organic synthesis and transformation reactions, such as alkylation, condensation, and dehydration [20,21,22,23]. Niwa et al. successfully measured the concentration of Brønsted and Lewis acid sites in SZ catalysts by using the ammonia infrared-mass spectroscopy/temperature-programmed desorption (IRMS–TPD) method [24]. The generation of strong Brønsted acid sites in the SZ catalysts is suggested to be primarily responsible for their high catalytic activity in *n*-heptane cracking. Furthermore, Huang et al. reported different ratios of Brønsted to Lewis acid sites in sulfated monoclinic and tetragonal zirconia phases—0.50 and 0.55, respectively [25]. However, the physicochemical properties of the SZ catalysts differ significantly from those of the synthesis method, calcination temperature, and sulfate ion precursors [26].

Here, we systematically investigated the effects of the synthetic parameters of SZ catalysts, such as calcination temperature and sulfate ion content, on the decomposition of formic acid. The acidity of SZ was characterized by the temperature-programmed desorption of *iso*-propanol (IPA-TPD) and pyridine-adsorbed infrared (Py-IR) spectroscopy. The relative contributions of the Brønsted and Lewis acid sites to the conversion of formic acid were compared.

## 2. Materials and Methods

### 2.1. Catalyst Preparation

Zr(OH)_4_ was synthesized by using zirconyl chloride (ZrOCl_2_∙8H_2_O, Kanto, 99%) and ammonia solution (28 wt.% NH_4_OH, SK Chemical). In a typical preparation, aqueous ammonia solution was added dropwise to a 0.5 M zirconyl chloride aqueous solution, under vigorous stirring, until a pH of 9.5 was reached. The resulting suspension was aged at 100 °C for 48 h and subsequently washed with distilled water until the pH of the filtrate reached 7.0, thereby confirming the complete removal of residual Cl^−^ ions. Finally, a Zr(OH)_4_ cake was recovered and dried at 100 °C for 24 h.

SZ catalysts with different sulfate ion contents (0–20 wt.%) were prepared by using the incipient wetness impregnation method. In a typical preparation, the desired amount of ammonium sulfate ((NH_4_)_2_SO_4_, Samchun, 99%) solution as a sulfating agent was added to the Zr(OH)_4_ powder and dried at 100 °C for 24 h. Finally, the sample was calcined at different temperatures in a range of 500–650 °C for 2 h (ramping rate of 2 °C min^−1^), under ambient air. The catalyst was denoted as *x*SZ(*y*), where *x* and *y* represent the sulfate content and calcination temperature, respectively. For comparison, pure ZrO_2_ was prepared by calcination of Zr(OH)_4_ at 600 °C for 2 h without an addition of (NH_4_)_2_SO_4_ and was denoted as 0SZ(600).

### 2.2. Characterization of Catalysts

Powder X-ray diffraction (XRD) analysis was performed on a D8 Discover (Bruker AXS, billericay, MA, USA), using Cu Kα radiation (*λ* = 0.15468 nm) in the 2*θ* range of 10–80° (scan rate = 0.009° s^−1^). The crystal structures of the catalysts were analyzed by using the Joint Committee on Powder Diffraction Standards (JCPDS) database. The specific surface area and pore size were measured by N_2_ sorption at −198 °C, using an ASAP2020 gas adsorption analyzer (Micromeritics, Norcross, GA, USA). Prior to the measurement, all samples were degassed at 250 °C for 4 h, under vacuum, and the surface area was determined by using the Brunauer–Emmett–Teller (BET) method from the relative pressure (P/P_0_), ranging from 0.05 to 0.20. The morphology of the catalysts was investigated by scanning electron microscopy (SEM), using an LEO-1530 microscope (Carl Zeiss, Oberkochen, Germany). The sulfate ion content was determined by using an elemental analyzer (Elementar vario MACRO cube, Langenselbold, Germany) and energy-dispersive X-ray spectroscopy (EDS, Carl Zeiss, Libra 120). Thermogravimetric analysis (TGA) was conducted on an SDT Q600 (TA Instruments) in temperatures ranging from 100 to 1000 °C (ramping rate of 10 °C min^−1^), under a flowing N_2_ (100 cm^3^ min^−1^) atmosphere. Fourier-transform infrared (FTIR) spectra were recorded on an IFS 66/S spectrometer (Bruker Optic Gmbh, Ettlingen, Germany) in the range of 400–4000 cm^−1^, with a resolution of 2 cm^−1^. The temperature-programmed decomposition (TPDE) experiment was performed by using a home-built apparatus with a mass spectrometer detector (Blazers QMS200, Nashua, NH, USA). Typically, ca. 0.05 g catalyst fixed in a U-shaped quartz reactor is heated from room temperature (RT) to 1000 °C (ramping rate of 10 °C min^−1^), under an Ar atmosphere. The IPA-TPD experiment was also conducted on the same apparatus as the TPDE, using a similar procedure. Here, a ca. 0.05 g of sample was pretreated at 300 °C for 1 h, under flowing Ar (30 cm^3^ min^−1^), exposed to 3% IPA (30 cm^3^ min^−1^, balanced with Ar) flow for 0.5 h at RT, and subsequently purged with flowing Ar (30 cm^3^ min^−1^) for 0.5 h to remove the physically adsorbed IPA on the catalyst’s surface. The IPA-TPD profile was measured by heating the sample from RT to 400 °C (ramping rate = 10 °C min^−1^), and the mass signal corresponding to *m/z* = 41 (C_3_H_5_) was recorded. Py-IR spectra were obtained on a Thermo Nicolet 6700 (Thermo Fisher Scientific, Waltham, MA, USA) by using self-supporting catalyst wafers of approximately 30 mg (1.3 cm diameter). Prior to the measurements, the catalyst wafers were pretreated under vacuum at 300 °C for 1 h inside a home-built IR cell with ZnSe windows, exposed to 64 μmol of pyridine at 150 °C, and then evacuated at the same temperature to remove the physisorbed pyridine. IR spectra were collected at 150 °C (32 scans with a resolution of 4 cm^−1^), and the concentrations of Brønsted and Lewis acid sites were calculated from the intensities of the IR bands at approximately 1550 and 1450 cm^−1^, respectively, using the molar extinction coefficients reported by Emeis [27]. X-ray photoelectron spectroscopy (XPS) was performed on a PHI Quantera-II (Ulvac-PHI, Chigasaki, Kanagawa, Japan) instrument with monochromatic Al-K*α* radiation (*hν* = 1486.6 eV). XPS data were calibrated by referencing the binding energy of adventitious carbon (C 1 s, 284.6 eV) as the standard. All spectral deconvolutions were performed by using Origin 9.0, a curve-fitting function.

### 2.3. Activity Test

The performance of the SZ catalyst was measured in a fixed-bed quartz reactor, under atmospheric pressure. Prior to the test, ca. 0.1 g of the catalyst was routinely activated under flowing Ar (100 cm^3^ min^−1^) at 400 °C for 1 h and then cooled to the reaction temperature (260 °C). Isothermal activity tests were performed by introducing 5% formic acid (balanced with Ar) at a total flow rate of 100 cm^3^ min^−1^. The reactor effluent was analyzed online by using a gas chromatograph (CP 3800, Varian) equipped with a Proapak Q column (1.8 m length and 1/8” o.d.) and a thermal conductivity detector (TCD). Formic acid conversion was calculated by using the following equation:$$\mathrm{Conversion}\;(\%)=\;\frac{F_{HCOOH\;in\;}-F_{HCOOH\;out\;}}{F_{HCOOH\;in}}\times 100$$
where *F_HCOOH in_* and *F_HCOOH out_* are the molar flow rates of formic acid at the inlet and outlet, respectively. The selectivity for CO was 100%, and CO_2_ formation was not observed for any of the SZ catalysts used in this study. The carbon balance over all catalysts employed in this study was 100%.

## 3. Results and Discussion

### 3.1. Physicochemical Properties of SZ Catalysts

Figure 1a shows the XRD patterns of the 5SZ(*y*) catalysts calcined at different temperatures (*y* = 500–650 °C). All the catalysts exhibited a tetragonal crystalline phase (JCPDS No. 50–1089). Furthermore, peaks corresponding to the monoclinic crystalline phase (JCPDS No. 37–1484) were not detected. Although the crystallinity of 5SZ(500) was very low owing to the insufficient temperature to crystallize ZrO_2_, a fully crystallized tetragonal phase was observed in the 5SZ(600) and 5SZ(650) catalysts, indicating that a temperature higher than 600 °C is required to completely crystallize the SZ catalysts. Ward and Ko also reported a similar temperature of 500 °C to crystallize 5 mol% SZ and found that the crystallization temperature of this material increased with sulfate loading [28]. However, the BET surface areas of the 5SZ catalysts continuously decreased from 234 to 145 m^2^ g^−1^ as the calcination temperature increased from 500 to 650 °C (Table 1). This surface-area loss was accompanied by an increase in particle size (3.8 → 7.0 nm) calculated by the Scherrer equation, representing the sintering of ZrO_2_ crystals during the crystallization process, which was further confirmed by SEM analysis (Appendix A). However, the pore sizes of the 5SZ(*y*) catalysts, as shown in Appendix A, were slightly increased from 8.3 to 8.7 nm, owing to the crystallized tetragonal phase [29,30]. From the EDS image of the 5SZ(600) catalyst, a uniform distribution of sulfate species on the surface of ZrO_2_ was identified (Appendix A). The sulfate ion contents in 5SZ(*y*), as measured by elemental analysis, decreased from 4.8 to 3.5 wt.% with an increase in the calcination temperature, owing to the partial decomposition of sulfate ion to SO_2_ and O_2_. The sulfate ion contents in the 5SZ(*y*) catalysts were matched with the weight losses calculated from the TGA experiments, as shown in Appendix A. In the TGA analysis, the weight loss below 600 °C corresponds to the desorption of physically adsorbed water molecules and dihydroxylation of surface hydroxyl groups, and that observed at higher temperatures (>600 °C) is attributed to the thermal decomposition of sulfate species [31,32]. The sulfate ion density, defined as the number of sulfate ions per nm^2^ on the surface of the 5SZ(*y*) catalysts, ranged from 1.3 to 1.5.

As shown in Figure 1b, the crystallinity of *x*SZ(600) catalysts steadily decreased as the sulfate ion loading (*x* = 0, 5, 10, 15, and 20) in SZ increased. However, XRD peaks other than those corresponding to the tetragonal phase were not observed. Notably, the BET surface areas of *x*SZ(600) catalysts showed a volcano-shaped distribution with respect to the sulfate ion content, exhibiting a maximum value (172 m^2^ g^−1^) at 10 wt.% loading. The decrease in the BET surface area for the *x*SZ(600) samples with *x* = 15 and 20 can be explained by the formation of polysulfate species, such as pyrosulfate, partially blocking the pores of ZrO_2_ [33,34]. Unlike that of 5SZ(y), the particle size and pore size of the *x*SZ(600) catalysts continuously decreased as the sulfate ion content increased (Appendix A); this outcome can be attributed to the decrease in the XRD peak intensity of the tetragonal phase, as shown in Figure 1. This retardation of crystallization by the addition of sulfate ions was also reported by Ward and Ko [28]. The sulfate ion contents in *x*SZ(600) catalysts increased from 4.0 to 12.6 wt.%, and the sulfate densities were proportionally increased from 1.5 to 5.6 SO_4_^2−^ ions per nm^2^ by adding sulfate species. The much higher weight loss (ca. 9.9 wt.%) observed for 20SZ(600) than for other catalysts shown in Appendix A also supported the presence of massive sulfate species in this catalyst. As reported by Katada et al., a monolayer of sulfate species on the ZrO_2_ surface was formed at 3.2 SO_4_^2−^ ions per nm^2^ by applying its kinetic diameter [35]. Bensitel et al. and Morterra et al. reported that the transition of isolated sulfate ions to polysulfate species occurs at sulfate ion densities higher than 1.5 SO_4_^2−^ ions per nm^2^ [36,37]. Thus, the formation of multilayer sulfate or polysulfate species on *x*SZ(600) catalysts can be expected for sulfate contents higher than 15 wt.% (*x* > 15).

The sulfate species on ZrO_2_ were characterized by tracing the mass signal of the ·SO_2_ fragment (*m/z* = 64) during the TPDE experiment (Figure 2 and Table 2). Deconvolution of the mass signals shown in Figure 2 generated three different peaks with respect to the decomposition temperatures: 700 (Peak I), 713 (Peak II), and 815 °C (Peak III). Although Peak III prevailed across samples, Peak I was observed for the 5SZ(*y*) samples calcined at relatively low temperatures (i.e., 5SZ(500) and 5SZ(550)), owing to the weak interaction of sulfate species with the amorphous ZrO_2_ domain. Furthermore, Glover et al. reported the low-temperature (600−700 °C) decomposition of sulfate species on the Zr(OH)_4_ surface [38]. As shown in Appendix A, the TPDE of the as-prepared 10SZ catalyst produced ·SO_2_ and a small number of ·NH (*m/z* = 15) fragments at 700 °C. Thus, the decomposition of ammonium pyrosulfate ((NH_4_)_2_S_2_O_7_), a major compound produced during the decomposition of (NH_4_)_2_SO_4_, cannot be ruled out [39]. However, these peaks completely disappeared for 10SZ(600), and the evolution of only the ·SO_2_ fragment at 850 °C was observed. The three representative conformations of the surface-bound sulfate species on ZrO_2_ are illustrated in Figure 1 [40,41]. Here, tridentate (Type I) and bridging bidentate (Type II) species were reported to be the most stable, decomposing at temperatures higher than 800 °C [42]. Considering that the Type I conformation is a dehydrated form of Type II, ·SO_2_ desorption at 815 °C can be attributed to the decomposition of Type I species [43]. As shown in Figure 2b, Peak II, which is centered at 713 °C, was observed in the samples with high sulfate content (>15 wt.%), that is, 15SZ(600) and 20SZ(600). This can be rationalized by the formation of new surface sulfoxy species (Type III in Figure 1) at high sulfate loadings, which are more easily decomposed in the lower temperature region. Rabee et al. observed the generation of an SO_2_ fragment at 670 °C for the SZ catalysts with high sulfate content owing to the decomposition of pyrosulfate anions (S_2_O_7_^2−^) [42].

The nature of the surface sulfate species on the SZ catalysts was further investigated by using FTIR spectroscopy (Figure 3). Except for 0SZ(600), all SZ catalysts exhibited IR absorption bands from 900 to 1300 cm^−1^, which are characteristic bands of surface sulfate species [44]. The bands at 1130 and 1225 cm^−1^ corresponded to the S=O vibrations in the bridging bidentate sulfate coordinated to Zr^4+^, while those at 995, 1030, and 1076 cm^−1^ corresponded to the S–O stretching vibrations [45,46]. In addition, the band observed at 1628 cm^−1^ was attributed to the bending vibrations of adsorbed water [47]. Notably, the intensity of the bands from S–O stretching vibrations (1030 and 1076 cm^−1^) for the 5SZ(*y*) samples was enhanced with an increase in the calcination temperature owing to the crystallization of the tetragonal ZrO_2_ phase (Figure 3a). This correlates well with the reduction of adsorbed water (1628 cm^−1^ in Figure 3a and 3700–3200 cm^−1^ in Appendix A) and amorphous ZrO_2_ phase in XRD patterns (Figure 1a). However, the IR band at ca. 813 cm^−1^, corresponding to Zr–O vibration, was observed in the 5SZ(600) and 5SZ(650) samples, owing to the removal of sulfate species from the ZrO_2_ surface by high-temperature calcination [48,49]. In contrast, the IR bands corresponding to S=O vibrations (1130 and 1225 cm^−1^) increased for *x*SZ(600) catalysts with an increase in the sulfate content. This is in line with the formation of Types II and III sulfate species derived from the TPDE experiments (Figure 2b). Unlike the continuous increase in S=O vibrations, the intensity of the band at 1628 cm^−1^ was maximized at 10SZ(600) and decreased with further increase in sulfate loading. The same trends were observed in the IR spectra of *x*SZ(600) samples in the hydroxyl region (Appendix A). This can be rationalized by the reaction of two adjacent bidentate species with pyrosulfate anions (2HSO_4_^−^ → S_2_O_7_^−^ + H_2_O), releasing a mole of water [50].

The changes in the chemical states of Zr and S in the SZ catalysts with respect to the calcination temperature and sulfate loading were characterized by XPS (Figure 4 and Table 3). The Zr 3d_5/2_ spectra of all catalysts showed two chemical states, namely from 181.9 to 182.5 eV (Peak I) and from 182.9 to 183.6 eV (Peak II), which were assigned to Zr–O and Zr–OH, respectively [51,52,53,54]. The S 2p_3/2_ peak centered at 169±0.1 eV was assigned to S^6+^ for SO_4_^2−^ [55,56]. The O 1s peak in Appendix A was deconvoluted into three peaks, namely 530.0–530.4 eV (Peak I), 531.6–532.2 eV (Peak II), and 533.0 eV (Peak III), arising from the oxygen in Zr–O, Zr–OH, and sulfate species, respectively [57,58]. Notably, the binding energies of Zr 3d_5/2_, S 2p_3/2_, and O 1s in the 5SZ(*y*) catalysts were almost unchanged with respect to the calcination temperature, representing identical chemical states of Zr, S, and O in these catalysts (Table 3). However, the relative proportion of Peak I (Zr–O) in the Zr 3d_5/2_ spectra of the 5SZ(*y*) catalysts increased from 66.7 to 73.5%, owing to the crystallization of amorphous ZrO_2_ to the tetragonal phase at high temperatures. In addition, in line with the reduction in sulfate content shown in Table 1, the intensity of S 2p_3/2_ in the 5SZ(*y*) catalysts was reduced with an increase in the calcination temperature. In the case of *x*SZ(600) catalysts, all binding energies of Zr 3d_5/2_, S 2p_3/2_, and O 1s were higher than those of 0SZ(600) and steadily increased with the increase in sulfate loading, owing to the strong interaction between the sulfate species and ZrO_2_ [59]. Notably, the relative proportion of Zr 3d_5/2_ and O 1s peaks centered at 183.2−183.4 eV and 531.7−531.9 eV, respectively, corresponding to Zr–OH species, was maximized on the catalyst with 10 wt.% sulfate loading (10SZ(600)). The highest proportion of Zr–OH on the 10SZ(600) catalyst was also correlated with the highest intensity of the 1628 cm^−1^ band in the FTIR spectrum (Figure 3b).

The acidic properties of the SZ catalysts were characterized by using IR spectroscopy with pyridine adsorption (Figure 5 and Table 2). The characteristic bands observed at 1444, 1580, and 1610 cm^−1^ correspond to the coordinatively adsorbed pyridine on Lewis acid sites, while the band at 1540 cm^−1^ corresponds to the pyridinium ion on Brønsted acid sites [60,61]. The concentration of acid sites on the catalysts was calculated by using the integrated area of the bands at 1444 and 1540 cm^−1^ for the Lewis and Brønsted acid sites, respectively, by applying the molar extinction coefficients reported by Emeis and normalized by S_BET_ values [27]. As shown in Table 2, although the densities (0.59−0.62 μmol m^−2^) of Lewis acid sites on 5SZ(*y*) catalysts did not vary considerably in the whole temperature region, those of Brønsted acid sites were linearly increased with calcination temperature up to 600 °C and decreased at the higher temperature. However, the distribution of Lewis acid as well as Brønsted acid densities on *x*SZ(600) catalysts was volcano-shaped with maxima at 0.78 and 0.66 μmol m^−2^, respectively, with 10 wt.% sulfate loading (10SZ(600)). A further increase in the sulfate ions on the SZ catalysts reduced the concentration of acid sites, owing to the transformation of Brønsted acidic bridging bidentate species (Type II) to pyrosulfate-like species, as evidenced by the formation of Peak II in Figure 2b. However, this transformation of Type II bridging bidentate to Type III surface sulfoxy species does not promote the Lewis acidity of catalysts. This is probably caused by the shielding of ZrO_2_ sites by excessive amounts of sulfoxy species and the reduction of surface areas of catalysts.

The acid strengths of the SZ catalysts were investigated by using IPA-TPD as a test reaction (Figure 6). Generally, the peak temperature (T_max, 41_) of ·C_3_H_5_ (*m/z* = 41) desorption originating from IPA dehydration is an indicator of the acid strength [11]. The absence of ·CH_3_CO (*m/z* = 43) fragment on 0SZ(600) during IPA-TPD represents the lack of base sites on this catalyst (Appendix A). As shown in Figure 6 and Table 2, the T_max, 41_ of 0SZ(600) was 235 °C, indicating its low acidity and low activity in IPA conversion. However, the lowest T_max, 41_ (126 °C) of ·C_3_H_5_ desorption was observed for 5SZ(600) among the 5SZ(*y*) catalysts, implying that this catalyst is highly acidic. This was in good agreement with the highest Brønsted density of 5SZ(600), as shown in Figure 5 and Table 2. Meanwhile, the T_max, 41_ of ·C_3_H_5_ desorption on the 5SZ(650) catalyst was slightly increased, implying a reduction in acidity. In the case of *x*SZ(600) catalysts, the lowest T_max, 41_ of C_3_H_5_ desorption was observed for the 10SZ(600) catalyst, which was also correlated with the highest acid densities of both Lewis and Brønsted acids (Figure 5). In line with the decrease in the acid densities for the 15SZ(600) and 20SZ(600) catalysts, an increase in of T_max, 41_ was observed for these catalysts, indicating their reduced acid strength.

### 3.2. HCOOH Decomposition

The catalytic performances of the SZ catalysts for the decomposition of formic acid at 260 °C were compared, as shown in Figure 7. Here, CO was the sole product of all the employed catalysts, with 100% selectivity. As shown in Figure 7a, although steady deactivation was observed for all the 5SZ(*y*) catalysts, the conversion of formic acid during the test period decreased in the following order: 5SZ(600) > 5SZ(650) > 5SZ(550) > 5SZ(500). This trend in formic acid conversion over the 5SZ(*y*) catalysts was exactly matched with T_max, 41_ in Table 2, indicating that the strength of the acid sites plays a decisive role in the dehydration of formic acid. As shown in Appendix A, 10SZ(600) exhibited the highest formic acid conversion among the 10SZ(*y*) catalysts. Thus, the optimal calcination temperature for the SZ catalysts was identified as 600 °C. For the *x*SZ(600) catalysts shown in Figure 7b, the conversion of formic acid was also well correlated with T_max, 41_ in Table 2 and decreased in the following order: 10SZ(600) > 15SZ(600) > 5SZ(600) ≥ 20SZ(600) > 0SZ(600). Unmodified 0SZ(600) exhibited very poor catalytic activity, with a conversion of less than 40%. Unlike the other *x*SZ(600) catalysts, the conversion of formic acid over 10SZ(600) and 15SZ(600) was stabilized after 1 h, and no significant deactivation was observed. The initial conversion of formic acid over *x*SZ(*y*) catalysts was plotted as T_max, 41_ and acid site density (Figure 8). As shown in Figure 8a, the conversion of formic acid over 5SZ(*y*) catalysts was inversely related to T_max, 41_ and varied with the density of Brønsted acid sites. However, it was less influenced by the density of the Lewis acid sites. In the case of *x*SZ(600) catalysts, the same relationship between formic acid conversion and Brønsted acidity was observed. The decrease in formic acid conversion on *x*SZ(600) catalysts with *x* > 15 corresponded to a decrease in the BET surface area and an increase in the pyrosulfate proportion (Table 1 and Figure 2). This is also in line with the reduction of Zr–OH species, which is closely related to the structure of bridging bidentate sulfate species, observed on XPS (Figure 4 and Appendix A) and IR (Figure 3). To understand the reaction mechanism of formic acid dehydration over the SZ catalyst, an on–off cycle test was conducted at 260 °C, using the most active 10SZ(600) catalyst (Figure 9). Notably, a sharp increase in the CO (*m/z* = 28) and ·HCO (*m/z* = 29) fragments was observed from the onset of the reaction, while the evolution of ·H_2_O (*m/z* = 18) was delayed by approximately 1 min. This represents the transformation of Type I tridentate sulfates to Type II bridging bidentate species (Brønsted acids) by reacting with the water produced during formic acid dehydration. The protonated formic acid on a strong Brønsted acid site liberates water molecules at 100−150 °C, and the remaining charged acyl moiety (O=CH)^+^ transforms into CO and surface OH group by interacting with nucleophilic oxygen in sulfate species [62]. In addition, after the formic acid injection was turned off, the intensities of the mass signals for ·CO and ·HCO decreased successively. However, the intensity of the mass signal for H_2_O slowly decreased and became invisible after 15 min, indicating strong adsorption of water on the SZ catalyst. This can be indirect evidence of Brønsted acid generation by the reaction of tridentate sulfates with the produced water.

## 4. Conclusions

Two series of SZ, prepared by varying the calcination temperature and sulfate loading, were applied as catalysts for the dehydration of formic acid to carbon monoxide. Different sulfate species on the SZ catalysts, such as tridentate, bridging bidentate, and pyrosulfate, were identified by using TPDE, FTIR, and XPS analyses. The acidic properties of the SZ catalysts measured by Py-IR and IPA-TPD were correlated with the catalytic activity for formic acid dehydration. The conversion of formic acid on the SZ catalysts was more dependent on the Brønsted acidity, which is represented by the acid density and T_max, 41_. The formation of pyrosulfate species on SZ catalysts with high sulfate loading (>15%) was found to have a detrimental effect on acidity and catalytic activity. The optimal calcination temperature and sulfate loading for the SZ catalysts with the highest CO yields were 600 °C and 10 wt.%, respectively. The overall results of this study suggest optimal preparation conditions for catalysts yielding high-purity CO from formic acid dehydration.

## Data Availability

Not applicable.

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
