# Peer review of "Production of H_2_-Free Carbon Monoxide from Formic Acid Dehydration: The Catalytic Role of Acid Sites in Sulfated Zirconia"

_nanomaterials, 2022, doi:10.3390/nano12173036_

Round 1

Reviewer 1 Report

This paper describes the preparation of modified Zirconia catalysts for the deomcposition of formic acid into carbon monoxide, a process with high industrial relevance. The modification of zirconia give materials with various degrees of sulfation, and a series of compounds are carefully characterised, and tested for catalytic activity.

This article is very well written, the presented data support the conclusions well, and it is certain to be of interest to the readers of Nanomaterials. 

My only very minor comment, is that I am not convinced that the BHJ method gives pore sizes with such accuracy that it can be concluded that there are significant differences in the pore size distributions of the different materials (data shown in Table 1). The authors could perhaps comment on this.

Other than that, I have no other comment. It is probably the first time that I have so few remarks to make on a submitted paper, and I congratulate the authors on a nice piece of work.

Author Response

  1. My only very minor comment, is that I am not convinced that the BHJ method gives pore sizes with such accuracy that it can be concluded that there are significant differences in the pore size distributions of the different materials (data shown in Table 1). The authors could perhaps comment on this.

A1: I appreciate this reviewer for the good comments on my manuscript. As reviewer mentioned, the pore size of catalysts cannot be determined accurately by BJH method. The pore size of catalysts in Table 1 was defined as the maximum of pore size distribution curve measured by BJH method. Here, the average pore diameter was not calculated owing to the interference by the artificial peak from tensile strength effect (TSE). Thus, the pore size in Table 1 represents the representative pore size of the catalysts not an accurate value. The values of pore size were presented to one decimal place in order to reflect the variation of pore size with respect to the calcination temperature and sulfate contents.

Reviewer 2 Report

This manuscript reported the formic acid decomposition over the sulfated zirconia catalysts. The effects of catalysts preparation conditions on the HCOOH decomposition activity and surface acidity were investigated intensively. A lot of meaningful results and conclusion have been given. I consider that it can be published in the journal of Nanomaterials. However, there are some issues that need to be discussed with the authors as following.   

1.     Just as reported in this paper, there are two pathways of HCOOH dehydration: 1) HCOOH ® CO + H2O; 2) HCOOH ® CO2 + H2. But in the reaction carried out by the authors, the CO was the sole production over all catalysts used. Why is the second reaction suppressed? What is the reaction mechanism in here? What is the carbon balance data for the chemical reaction?

2.   There are three surface sulfate species on the SZ catalysts. It is best for the author to give intensive discussion of relationship beteem the species and surface Lewis and B acidic.

3.     On the surface of the SZ catalysts, the Zr-OH species are also observed. Does it have effects on the HCOOH dehydration reactivity over the SZ catalysts?

Author Response

  1. Just as reported in this paper, there are two pathways of HCOOH dehydration: 1) HCOOH ® CO + H2O; 2) HCOOH ® CO2 + H2. But in the reaction carried out by the authors, the CO was the sole production over all catalysts used. Why is the second reaction suppressed? What is the reaction mechanism in here? What is the carbon balance data for the chemical reaction?

A1: I appreciate this reviewer for the good comments on my manuscript. As we mentioned in the introduction part, HCOOH decomposition could proceeds via two different routes: dehydration (CO + H2O) over acid catalysts, and dehydrogenation (H2 + CO2) over metal or base catalysts. Although IPA-TPD in Figure 6 can characterize both acidic and basic properties of catalyst by detecting ·C3H5 (m/z = 41) and ·CH3CO (m/z = 43) fragments, respectively, the latter fragment was not observed over all catalysts representing the absence of base sites. Thus, the dehydrogenation of formic acid producing H2 and CO2 was suppressed. We added a sentence as highlighted in red “The absence of ·CH3CO (m/z = 43) fragment on 0SZ(600) during IPA-TPD represents the lack of base sites on this catalyst (Figure S9).” in the Results and discussion section of the revised manuscript. In addition, Figure S9 with caption “Figure S9. Evolution of ·CH3CO (m/z = 43) and ·C3H5 (m/z = 41) in mass spectra as a function of temperature during IPA-TPD of 0SZ(600).” was also added in the Supplementary Materials.

Figure S9. Evolution of ·CH3CO (m/z = 43) and ·C3H5 (m/z = 41) in mass spectra as a function of temperature during IPA-TPD of 0SZ(600).

The reaction mechanism of formic acid dehydration over Brønsted acid sites was further described in Results and discussion section as highlighted in red “The protonated formic acid on a strong Brønsted acid site liberates water molecule at 100−150 °C, and the remaining charged acyl moiety (O=CH)+ transforms into CO and surface OH group by interacting with nucleophilic oxygen in sulfate species [62].” and the corresponding reference also added in Reference section “[62] Popova, G. Ya.; Zakharov, I.I.; Andrushkevich, T.V. Mechanism of formic acid decomposition on P-Mo heteropolyacid. React. Kinet. Catal. Lett. 1999, 66, 251–256. https://doi.org/10.1007/BF02475798.” in our revised manuscript. 

The conversion and selectivity was calculated based on the molar fraction of HCOOH, CO, and CO2. As CO2 was not detected over all catalysts, the carbon balance was simply calculated by the carbon number of HCOOH and CO and obtained as 100%. To more clearly describe this, we added a sentence as highlighted in red “The carbon balance over all catalysts employed in this study was 100%.” in the Experimental section of the revised manuscript.

  1. There are three surface sulfate species on the SZ catalysts. It is best for the author to give intensive discussion of relationship beteem the species and surface Lewis and B acidic.

A2: I appreciate this reviewer for the helpful comments on my manuscript. To more clarify the relationship between the sulfate species and Lewis and Brønsted acidity, the additional description as highlighted in red “However, this transformation of type II bridging bidentate to type III surface sulfoxy species does not promote the Lewis acidity of catalysts. This is probably caused by the shielding of ZrO2 sites by excessive amounts of sulfoxy species and the reduction of surface areas of catalysts.” was added in the Results and discussion section of the revised manuscript.

  1. On the surface of the SZ catalysts, the Zr-OH species are also observed. Does it have effects on the HCOOH dehydration reactivity over the SZ catalysts?

A3: The highest proportion of the surface Zr-OH species measured by XPS analysis (Figure 4 and Figure S8) over 10SZ(600) was well correlated to the highest conversion of this catalyst  shown in Figure 7b. As shown in Scheme 1, the hydroxyl species are only observed on type II bidentate sulfate species, which are the sites most active for dehydration of formic acid. To more clearly describe this, we added a sentence as highlighted in red “This is also in line with the reduction of Zr–OH species, which is closely related to the structure of bridging bidentate sulfate species, observed on XPS (Figure 4 and S8) and IR (Figure 3).” in the Results and discussion section of the revised manuscript.

Reviewer 3 Report

Publish after minor spell check.

One major comment is that 5% of formic acid in stream of N2 is not very applicable. However, as a first step to optimization of conditions, it's acceptable.

Author Response

  1. One major comment is that 5% of formic acid in stream of N2 is not very applicable. However, as a first step to optimization of conditions, it's acceptable.

    A1: I appreciate this reviewer for the good comments on my manuscript. As this reviewer recognized, the purpose of this study is a fundamental understanding of correlation between the synthetic parameters of catalyst preparation and the acidic and catalytic properties of SZ catalysts. We believe that the overall results of this study can be used as a guideline for the development of practical catalysts applicable for high purity CO production process.